# Montelukast Nanocrystals for Transdermal Delivery with Improved Chemical Stability

**DOI:** 10.3390/pharmaceutics12010018

**Published:** 2019-12-23

**Authors:** Sung Hyun Im, Hoe Taek Jung, Myoung Jin Ho, Jeong Eun Lee, Hyung Tae Kim, Dong Yoon Kim, Hyo Chun Lee, Yong Seok Choi, Myung Joo Kang

**Affiliations:** College of Pharmacy, Dankook University, 119, Dandae-ro, Dongnam-gu, Cheonan-si, Chungcheongnam-do 31116, Korea; spayy@naver.com (S.H.I.); jht1219@naver.com (H.T.J.); butable@gmail.com (M.J.H.); jjingki_timo@naver.com (J.E.L.); httkk@naver.com (H.T.K.); skykimdy@naver.com (D.Y.K.); gycjs2015@naver.com (H.C.L.)

**Keywords:** montelukast, nanocrystal suspension, transdermal delivery, nanocrystal hydrogel, photo-stability

## Abstract

A novel nanocrystal system of montelukast (MTK) was designed to improve the transdermal delivery, while ensuring chemical stability of the labile compound. MTK nanocrystal suspension was fabricated using acid-base neutralization and ultra-sonication technique and was characterized as follows: approximately 100 nm in size, globular shape, and amorphous state. The embedding of MTK nanocrystals into xanthan gum-based hydrogel caused little changes in the size, shape, and crystalline state of the nanocrystal. The in vitro drug release profile from the nanocrystal hydrogel was comparable to that of the conventional hydrogel because of the rapid dissolution pattern of the drug nanocrystals. The drug degradation under visible exposure (400–800 nm, 600,000 lux·h) was markedly reduced in case of nanocrystal hydrogel, yielding only 30% and 50% amount of *cis*-isomer and sulfoxide as the major degradation products, as compared to those of drug alkaline solution. Moreover, there was no marked pharmacokinetic difference between the nanocrystal and the conventional hydrogels, exhibiting equivalent extent and rate of drug absorption after topical administration in rats. Therefore, this novel nanocrystal system can be a potent tool for transdermal delivery of MTK in the treatment of chronic asthma or seasonal allergies, with better patient compliance, especially in children and elderly.

## 1. Introduction

Montelukast (MTK) is a selective leukotriene receptor antagonist that has been commonly prescribed in the treatment of chronic asthma and symptoms of seasonal allergies in children and adults [1,2,3]. Moreover, recently, MTK is clinically being investigated as a novel medication in the treatment of Alzheimer’s disease in elderly people [4]. In preclinical studies, the leukotriene antagonist was found effective in stroke models and was beneficial in improving cognition in older rats, owing to its anti-inflammatory and neuroprotective effects [5,6]. Considering its frequent prescription to children in the age group of 6–24 months [7] and potential dosing in elderly patients with difficulty in swallowing, the transdermal dosage form can be an alternative for better patient compliance. However, it is quite challenging to formulate the therapeutic agent into a transdermal dosage form because of its poor aqueous solubility (0.2–0.5 μg/mL in water at 25 °C) [8,9] and chemical instability [10,11]. MTK is extremely susceptible to light- or oxygen-induced degradation, and mainly breaks down into sulfoxide and *cis*-isomer by the oxidation of the mercapto group and by isomerization of the double bond, respectively. Al Omari et al. (2007) reported that drug content in the methanolic solution was drastically decreased to 70% upon UV exposure for 15 min, producing a substantial amount of *cis*-isomer [11]. Moreover, the penetration of this lipophilic compound (log *P* > 8.0) across the relevant skin layer and the absorption into the bloodstream are strictly restricted by the “brick and mortar” structure of stratum corneum [12,13,14,15,16].

Recently, drug nanocrystallization technique has emerged as a promising topical and/or transdermal delivery modality for lipophilic pharmaceuticals and/or cosmetic substances [17,18]. Drug nanocrystals are colloidal dispersions of sub-micronized drug particles stabilized by minimal polymeric and/or amphiphilic stabilizers in a continuous phase [19,20]. Compared to micronized preparations, drug nanocrystals exhibit markedly increased saturation solubility and dissolution rate with increasing surface-to-volume ratio. The increased saturation solubility in the formula escalates the concentration gradient between the topical formulation and the relevant skin layer, leading to an elevated diffusive flux value [21,22]. Nanocrystals below 500 nm in size have been reported to penetrate the skin through the hair follicles, and subsequently being absorbed by the surrounding follicular epithelium [23]. Moreover, drug crystal system can provide higher chemical stability as compared to the conventional drug-solubilized formulations by minimizing the exposure to light, oxygen, and moisture at the molecular level [24]. Lai et al. have revealed that the half-life of tretinoin, a photo-liable anti-wrinkling agent was 7.5-fold longer with nanosuspension, as compared to drug methanolic solution [25].

Herein, the aims of this study were to design nanocrystal systems of MTK, and to evaluate the physical characteristics, chemical stability, and in vivo pharmacokinetic profiles in rats. Drug nanocrystal suspensions possessing uniform size distribution and sound dispersibility were prepared by acid-base neutralization and ultra-sonication technique, by optimizing the types and concentrations of the stabilizer, the drug concentrations in the solvent, and the sonication conditions. The prepared nanocrystals were embedded into the xanthan gum-based hydrogel and were characterized in terms of their crystal size, morphology, crystallinity, and in vitro dissolution profile. The drug stability of MTK nanocrystal system under light exposure was evaluated and compared with the drug solution or conventional hydrogel. Moreover, in vivo pharmacokinetic profile of MTK following topical application of nanocrystal hydrogel was evaluated in comparison to that of the conventional hydrogel in rats.

## 2. Materials and Methods

### 2.1. Materials

The sodium salt form of MTK was obtained from Kyongbo Pharm. (Asan, Chungnam, Korea). Polyvinylpyrrolidone K30 (PVP K30), Kollidon/vinylpyrrolidone-vinyl acetate (VA64) copolymer, Poloxamer-188, Poloxamer-407, Kolliphor RH40 (polyoxyl 40 hydrogenated castor oil), Kolliphor EL (polyoxyl castor oil), Kolliphor HS 15 (polyoxyl 15 hydroxystearate) were kindly provided by BASF Co. (Ludwigshafen, Germany). Hypromellose 2910 (HPMC 2910), and trifluoroacetic acid were purchased from Sigma Chemical Co. (St. Louis, MO, USA). Polysorbate 20 and polysorbate 80 were obtained from Masung & Co. Ltd. (Seoul, Korea). Pharma grade xanthan gum was purchased from Arthur Branwell Co. Ltd. (Essex, UK). Essex EssexLactic acid and sodium hydroxide standard solution (1 N) were acquired from Daejung chemical & metal Co. Ltd. (Gyeonggi-do, Korea). Sunset Yellow FCF (disodium 2-hydroxy-1-(4-sulfonatophenylazo)naphthalene-6-sulfonate) was obtained from Bolak Co. Ltd. (Gyeonggi-do, Korea). Acetonitrile (ACN) and methanol of HPLC (High-performance liquid chromatography) grade were provided by J.T. Baker (Phillipsburg, NJ, USA). All other reagents were analytical grade and were used without further purification.

### 2.2. Preparation of MTK Nanocrystal Suspension and Hydrogel

Drug nanocrystal suspension was fabricated using acid-neutralization method as reported previously [26,27], with slight modification. At first, drug alkaline solution (5–20 mg/mL as MTK) was prepared by dissolving sodium salt of MTK powder (62.4–249.6 mg) in 10 mM NaOH solution (4 mL, pH 12). Different kinds of stabilizers (PVP K30, Kollidon VA64, Poloxamer-188, Poloxamer-407, Kolliphor RH40, Kolliphor EL, Kolliphor HS 15, HPMC 2910, polysorbate 20, and polysorbate 80; Table 1) were separately dissolved in 60 mM lactic acid solution in the concentration ranges from 0.05–2.0 (%, *w*/*v*). The drug alkaline solution was then added to the stabilizer-containing acidic solution at the rate of 2 mL/min, under ultra-sonication procedure. The probe-type ultra-sonicator (Vibracell VC-505, Sonics & Materials, Newtown, CT, USA) equipped with a 1/2-inch (13 mm) probe was placed into the acid solution and sonicated under different amplitudes (11–33 watts) for 3 min, with different cycles (1–3 times). To prevent the temperature elevation, the samples were located inside the ice bath during sonication procedure. The acidity of the MTK nanosuspensions was adjusted to pH 4.0–5.0, to suspend the weak-acid compound in the aqueous vehicle. MTK nanosuspensions were then stored at room temperature for further experiments, under light protection.

MTK nanocrystal-loaded hydrogel was prepared by adding xanthan gum (1.0%, *w*/*v*) to the drug nanocrystal suspension and then stirring for overnight at 700 rpm (IKA RCT basic, Staufen, Germany). Prepared hydrogels were incubated overnight at room temperature to remove the air bubbles.

### 2.3. Preparation of Drug Solution and Conventional Hydrogel

Drug solution (10 mg/mL as MTK) was prepared by adding MTK-Na powder (124.8 mg) to 10 mM NaOH solution (12 mL) and stirred for 10 min at room temperature. The pH of the solution was adjusted to 11.5–12.0, to completely dissolve the weak-acid compound. To formulate conventional hydrogel, 120 mg of xanthan gum powder (1.0% *w*/*v*) was added to 12 mL of drug solution and stirred at 700 rpm for 12 h.

### 2.4. Morphological and Physicochemical Characterization of MTK Nanocrystal Formulations

#### 2.4.1. Particle Size and Zeta Potential of MTK Nanocrystals

Average particle size and polydispersity index (PDI) value, a measure of the width of the size distribution of MTK nanocrystal suspension were determined using a Zetasizer Nano dynamic light scattering particle size analyzer (Marlvern Instrument, Worcestershire, UK) [28,29,30]. Each sample was loaded onto disposable cells with no additional dilution at a scattering angle of 90°. The zeta potential of the MTK nanocrystals (approximately pH 4.0) was also estimated using a Zetasizer Nano at 25 °C. Samples (100 μL) were loaded into the capillary cell after 10-fold dilution with distilled water, and 20 runs were performed for each measurement. All measurements were carried out in triplicates at 25 °C.

#### 2.4.2. Morphology of MTK Nanocrystals

The morphology of MTK nanocrystals in the aqueous suspension or hydrogel was observed using a transmission electron microscopy (TEM, Tecnai F20 G2, FEI, Hillsboro, OR, USA). Approximately 50 μL of sample was loaded onto the copper grid and was gently blown up for 20 min, to diminish aqueous vehicle. Samples were then fixed on sample holder and observed under an accelerating voltage of 80 kV.

#### 2.4.3. X-ray Powder Diffraction Analysis

The crystalline state of drug powder, nanocrystal suspension, and nanocrystal-embedded hydrogel were analyzed using an X-ray diffractometer (XRD, Model Ultima IV, Rigaku, Japan) with Cu Kα radiation generated at 30 mA and 40 kV [31]. To remove the aqueous vehicle and thickening agent from the preparations, each sample was 10-fold diluted with distilled water and was centrifuged at 3500 g for 10 min, leading to the settling of the drug nanocrystals. Afterward, the collected MTK nanocrystals were dried in the oven at 40 °C for 12 h. Samples were loaded on the glass plate and the diffraction pattern over a 2*θ* range of 5°–45° was determined using a step size of 0.02°. Scan speed was fixed to 1.0 s/step.

#### 2.4.4. Thermal Analysis

The thermal behaviors of the drug powder or dried nanocrystal formulations were analyzed using differential scanning calorimeter (DSC, Model DSC 50, Shimadzu, Japan) [32,33]. Nanocrystals were collected by dilution, ultra-centrifugation, and subsequent drying processes as described in Section 2.4.3. Each sample (about 2 mg) was put in an aluminum pan and heated at the rate of 10 °C/min over the temperature range of 0–200 °C. An empty aluminum pan was used as reference.

#### 2.4.5. Drug Content in the Nanocrystal Suspension

The amount of MTK suspended in the preparations was determined by HPLC assay. Nanocrystal suspension (1 mL) was ultra-centrifuged at 13,000 rpm for 10 min to settle the nanocrystals in the aqueous vehicle. Methanol was then added to the precipitate and vortexed for 30 min to dissolve the drug nanocrystals. The concentration of MTK in the organic solvent or in the supernatant was analyzed using Waters HPLC system that is composed of a pump (Model 515 pump), a UV–VIS (ultraviolet–visible) detector (Model 486), and an autosampler (Model 717 plus). The mobile phase comprised acetonitrile and distilled water at a volume ratio of 4:6 with 0.15% *v*/*v* of trifluoroacetic acid, was flowed through the reversed-phase C18 column (4.6 mm × 50 mm, 1.8 µm, Agilent, Santa Clara, CA, USA) at a flow rate of 1.2 mL/min. The column temperature was set to 25 °C. A 20 µL aliquot was injected and the column eluent was monitored at a wavelength of 238 nm. The sharp peak of MTK was detected at 10.7 min. The least-square linear regression was linear in the MTK range from 1.0–100 μg/mL, with a coefficient of determination (*r*2) value of 0.997.

### 2.5. In Vitro Dissolution Profiles of MTK Nanocrystal Formulations

In vitro dissolution profile of MTK from the nanocrystal formulations (suspension or hydrogel) was comparatively evaluated with the conventional preparations (drug solution or conventional hydrogel) under sink condition, according to USP II paddle method (Model DT 720, Erweka). To maintain sink condition during the release test, 0.5% *w/v* of sodium lauryl sulfate was dissolved in 10 mM phosphate buffered saline (pH 7.4) as the solubilizing agent. Each formulation containing 5 mg of MTK were added to 500 mL of dissolution media kept at 37 °C ± 0.5 °C and stirred at 50 rpm. At predetermined times (0, 0.2, 0.5, 1.0, 1.5, 2.0, 3.0, 4.0, 5.0, and 6.0 h), the dissolution media was withdrawn and replaced with equal volume of pre-warmed media. Withdrawn samples were then centrifuged at 13,000 rpm for 5 min to remove undissolved materials, including the drug nanocrystals. The supernatant was then 2-fold diluted with ACN and analyzed using HPLC as described earlier.

### 2.6. Photo-Stability of MTK Nanocrystal Formulations

The photo-stability of MTK nanocrystal formulations was comparatively evaluated with that of the conventional preparations by determining the drug remaining and degradation product formation, under stress conditions. For photo-stability study, samples containing Sunset Yellow FCF, as photo-stabilizer, at a concentration of 0.2% *w/v* were additionally prepared. A total of eight samples (nanocrystal suspension, nanocrystal hydrogel, drug alkaline solution, and conventional hydrogel with or without Sunset Yellow FCF) were placed into the scintillation vials and were kept in a DYX 500A solar simulator (DY-Tech, Seoul. Korea) under exposure to simulated sunlight, at a wavelength of 400–800 nm (8 h, 600,000 lux·h, 25 °C). At predetermined times, samples were withdrawn and then diluted with methanol. The drug remaining in each sample was then analyzed using HPLC as described earlier.

Additionally, the two main degradation products (*cis*-isomer and sulfoxide) of MTK in each sample were quantified by the area percentage method [34]. Each sample was diluted with methanol at a concentration of 100 μg/mL and were analyzed by HPLC gradient method. Mobile phase A was 0.15% trifluoroacetic acid-containing distilled water, and mobile phase B was 0.15% trifluoroacetic acid-containing acetonitrile. The mobile phase was passed into the reversed-phase C18 column (4.6 mm × 50 mm, 1.8 µm, packing L11, Agilent) following the gradient condition; 60% A phase for first 3 min, linear decrease to 49% over 16 min, return to 60% at 16.1 min, and maintenance over 5 min. Other parameters such flow rate, column temperature, and wavelength were identical to those of the drug content analysis method. Retention times of *cis*-isomer and sulfoxide was 5.7 min and 8.2 min, respectively.

### 2.7. In Vivo Transdermal Delivery of MTK Nanocrystal Formulations

#### 2.7.1. Animals and Experimental Protocols

In vivo pharmacokinetic study of MTK-loaded nanocrystal preparation was carried out in the healthy rats, after approval from Institutional Animal Care and Use Committee (IACUC) of Dankook University (Cheonan, Korea) (DKU-19-032, 8th October 2019). Six-week-old male Sprague-Dawley rats (SD rats, 250 ± 20 g) obtained from Samtako (Kyungki-do, Korea) were acclimated for at least 3 days, with free access to tap water and standardized chow. The rats were then divided into two groups (*n* = 6 per each group) and the hair on the dorsal region was removed by hair clippers without damaging the skin, prior to topical application. MTK nanocrystal hydrogel or conventional hydrogel were applied to the dorsal skin (20 cm^2^) at a dose of 25 mg/kg. To avoid the drying of the nanocrystal or conventional hydrogels, a thin film made from polyurethane and acrylamide (Tegaderm^®^, 10 × 12 cm) was attached onto the application site. At predetermined time intervals (0, 0.5, 1, 2, 3, 4, 5, 8, 12, and 24 h), the blood samples (about 0.2 mL) were withdrawn from the submandibular vein using 28 G heparinized syringe. At 6 h post-administration, rats were allowed to freely access to water and standardized chow. During the experimental period (24 h), individual rats were monitored for their changes in skin and fur, behavior pattern, morbidity, and mortality. Blood samples were centrifuged at 13,000 rpm for 5 min and stored at −70 °C until analysis. The drug concentration in the plasma was determined by the high-performance liquid chromatography and tandem mass spectrometry (LC–MS/MS) method described below.

#### 2.7.2. LC/MS-MS Analysis and Calculation of Pharmacokinetic Parameters

The MTK levels in rat plasma were determined using the liquid chromatography and multiple reaction monitoring (LC-MRM) method [35]. Briefly, 20 μL of thawed rat plasma was mixed with 500 μL of the extraction solution (25% (*v*/*v*) of dichloromethane and 75% (*v*/*v*) of ethyl acetate), including zafirlukast (the internal standard, IS, 60 ng/mL) and vortexed for 60 s. After the centrifugation at 12,000 g and 4 °C, the whole supernatant was dried under nitrogen stream. Then, the residue was reconstituted with 100 μL of methanol and the final solution was analyzed through the LC-MS/MS system, composed of Nexera X2 UHPLC and LCMS-8050 mass spectrometer (Shimadzu, Tokyo, Japan). For the interface between LC and MS, electrospray ionization in positive ion mode was employed. The separation of a sample was performed with Luna C18 column (2.0 × 150 mm, 5 μm, Phenomenex, Torrance, CA, USA). The two different kinds of mobile phases, 5 mmol/l ammonium formate in water (A) and methanol (B), were used with the gradient mobile phase program. The flow rate, the autosampler temperature, the volume of a sample injected, the column oven temperature, and the total analysis time were 0.25 mL/min, 4 °C, 5 μL, 40 °C, and 7 min, respectively. For the purpose of MTK quantitation, the MRM transition (screening transition) of 586.35 *m*/*z* (precursor ion)/422.30 *m*/*z* (product ion)/−25.0 V (collision energy), was employed. Additional transitions representing MTK (confirmatory transitions), 586.35 *m*/*z*/278.15 *m*/*z*/−35.0 V, and 586.35 *m*/*z*/440.25 *m*/*z*/−23.0 V were monitored for the purpose of confirming the identity of resulting transition peaks. For IS, 575.21 *m*/*z*/337.20 *m*/*z*/−22.0 V was employed as its screening transition, and 575.21 *m*/*z*/319.20 *m*/*z*/−33.0 V and 575.21 *m*/*z*/464.25 *m*/*z*/−14.0 V were used as its confirmatory transitions. Additional parameters for LC-MS/MS were as follows: the nebulizing gas flow of 3 L/min, the heating gas flow of 10 L/min, the interface temperature of 300 °C, the desolvation line temperature of 250 °C, and the heating block temperature of 400 °C. All LC-MRM data were acquired and analyzed using Lab Solutions software (version 5.93, Shimadzu, Kyoto, Japan). For the MTK concentrations, the screening transition peak area ratio of MTK to IS from a sample was compared with the calibration curve built from those of matrix-matched standard solutions.

From the pharmacokinetic data, pharmacokinetic parameters, such as area under the plasma concentration versus time curve (AUC) and terminal half-life (*T*_1/2_) were calculated using non-compartmental pharmacokinetics by using Winnonlin software (Version 5.2.1, Pharsight Co. Mountain View, CA, USA). Maximum plasma concentration (*C*_max_) and the time needed to reach the maximum plasma concentration (*T*_max_) were determined directly from the mean plasma concentration-time profile.

### 2.8. Statistical Analysis

Each experiment was performed at least three times and the data are presented as the mean ± standard deviation (SD). The statistical significance was determined using one-way analysis of variance (ANOVA) test and was considered to be significant at *p* < 0.05, unless otherwise indicated.

## 3. Results and Discussion

### 3.1. Selection of Steric Stabilizer of MTK Nanocrystal Suspension

The MTK free acid nanocrystals dispersed in aqueous media with different steric stabilizers was fabricated using the acid-base neutralization and the ultra-sonication techniques. In our preliminary study, anti-solvent precipitation provided more uniform drug crystal size as compared to the top-down methods, including the bead-milling process, and thus, was selected for further preparation of the MKT nanocrystal suspension (data not shown). In this method, the weak-acid MTK compound dissolved in the alkaline solution (10 mM NaOH solution) was added dropwise to the stabilizer-containing acidic solution (10 mM lactic acid solution). The drug solubility was drastically diminished in the acidic condition, instigating a rapid drug re-crystallization process [27]. The drug nanocrystals formed by the nucleation and crystal growth process were then ultra-sonicated to decrease the crystal size and uniformly disperse the nanocrystals using steric stabilizers.

In formulating drug nanosuspension, the types of dispersants and the interactions with the drug nanocrystals highly influence the colloidal stability in the vehicle [21]. Thus, different stabilizers, including hydrophilic polymers and surfactants were screened to formulate MTK nanosuspension based on the aspects of crystal size, homogeneity, and dispersibility. The concentration of stabilizer for screening test was set to 0.5% *w/v* through preliminary experiment (data now shown). When the stabilizer was not included, irregular drug crystals over 1.6 μm in size were formed (Table 1). The drug crystals with no stabilizer were thermodynamically unstable, forming agglomeration and/or precipitates after 7 days of storage under accelerated conditions (40 °C). The addition of steric stabilizers, such as HPMC-2910, Poloxamer-188 and 407, Polysorbate 20 and 80, Kolliphor RH40 and EL, and Solutol HS 15, markedly decreased the crystal size below 500 nm, by hindering the interaction of dissolved drug molecules with crystal surfaces. However, these surface stabilizers could not provide even dispersion of the MTK nanocrystals in aqueous vehicle, forming the drug aggregates after a week-long storage at 40 °C. On the other hand, when PVP K30 or Kollidon VA64 was dissolved in distilled water at a concentration of 1.0% *w*/*v*, MTK nanocrystals were physically stable with excellent re-dispersibility in the vehicle. These hydrophilic hydrogen-bonding acceptor polymers might be adsorbed on the surface of MTK nanocrystals by hydrogen bond and/or van der Waals interaction, and predominantly contribute to disperse the hydrophobic nanocrystals in the aqueous media, with no aggregation [36,37]. Out of the two hydrophilic polymers, PVP K30 polymer was selected for further preparation of MTK nanocrystal suspension, as it provided smaller and more uniform nanocrystals as compared to the VA64 polymer.

### 3.2. Effect of Process Parameters on Size and Homogeneity of MTK Nanocrystal Suspension

The effect of formulation variables, such as sonication powder and the number of sonication cycles, the drug and the PVP polymer concentrations in the preparation process, on crystal size and homogeneity is presented in Figure 1. Four formulation variables were set from the previous report [38] and our preliminary experiment. As expected [38], as the sonication intensity increased from 13–33 Watts, the drug crystal size markedly decreased, providing nano-sized drug crystals below 150 nm with a sonication power of 33 Watts (Figure 1a). Drug nanocrystal size was also decreased as the number of sonication cycles were increased, and they reached a plateau with three cycles (total 6 min), with a particle size below 150 nm (Figure 1b). The drug concentration in the nanosuspension was set to 10 mg/mL, with narrower size distribution (PDI value of 0.2) (Figure 1c). Under the arranged sonication condition, the MTK crystal size was effectively adjusted, by the concentration of PVP K30 polymer in the aqueous vehicle. When the concentration was increased from 0.05%–0.5%, the nanocrystal size was accordingly decreased from 900–150 nm (Figure 1d) as the polymeric stabilizer covered the surface of nanocrystals and thus, preventing and/or retarding drug crystal growth. Taken together, the optimized MTK nanocrystal suspension was prepared with the sonication power of 33 Watts, 6 min of sonication time, 10 mg/mL of drug concentration, and 0.5% *w*/*v* of PVP K30 polymer concentration.

### 3.3. Morphological and Physicochemical Characteristics of MTK Nanocrystal Suspension and Hydrogel

The optimized MTK nanocrystal suspension prepared with 0.5% *w*/*v* of PVP K30 polymer were further characterized in aspects of morphology, particle size, zeta potential, drug content, and drug crystallinity. The morphological features of the drug nanocrystals suspended in the aqueous vehicle or hydrogel matrix were scrutinized using TEM. In both aqueous suspension and hydrogel, MTK nanocrystals prepared by anti-solvent precipitation and ultra-sonication techniques were spherical and/or elliptical, and with homogeneous and smooth surface (Figure 2A,B). Dimensions of the novel MTK nanocrystals were uniform in the range of 100–200 nm, with no marked differences in the nanocrystal size in both formulas. The particle size analysis also revealed that MTK nanocrystals with median diameter of 102.3 nm were effectively prepared with PDI value < 0.3 (Table 2). The particle size of novel nanocrystal system was supposed to be appropriate for transdermal delivery [23,39,40]. In a previous report, nanocrystals with a size range of 200–400 nm offered enhanced permeability across the skin and mucosal membranes by providing an enhancement in the saturation solubility, and consequently, facilitated dissolution rate with reduced diffusional distance [39,40]. Moreover, nanocrystals with an appropriate size (approximately 700 nm) can deposit into these shunts, which act as a depot from which the drug can diffuse into the surrounding cells for extended release [23]. Pireddu et al. reported that 280 nm-sized nanocrystal formulation provided a higher accumulation of diclofenac in the skin as compared to both the coarse suspensions and the commercial formulation in ex vivo experiments [39].

Drug content analysis in suspension revealed that over 99.5% of MTK was suspended as solid-state in the formations, due to poor solubility of MTK, a weak-acid compound, in acidic environment (pH 4.1) (Table 2). The zeta potential of drug nanocrystals stabilized with PVP K30 polymer was neutral (−3.6 mV) (Table 2). The crystalline state of MTK nanocrystals dispersed in the aqueous media or hydrogel matrix was further evaluated by comparing the diffraction spectrum or thermal behavior of MTK nanocrystals with those of the MTK-Na, and MTK free acid powders (Figure 2C,D). The X-ray diffraction spectrums of the dried MTK nanocrystal suspension and hydrogel were analogous to those of MTK-Na, and MTK free acid powders, presenting no distinctive diffraction peaks over 2*θ* range of 5°–45° (Figure 2C). These results denote that the raw materials used in our study existed in an amorphous state and the crystalline form of the leukotriene antagonist did not alter during the nanocrystalization and subsequent gelation processes, retaining intrinsic amorphous form in both nanocrystal formulations. The drug crystalline state was further evaluated by analyzing thermal behavior of the nanocrystal formulas using the DSC measurement (Figure 2D). The DSC curves of raw materials showed broad endothermic peaks between 50–60 °C with no sharp peaks. These thermal patterns are consistent with previous reports showing that MTK powder is in an amorphous state, possessing glass transition temperature between 50 °C and 60 °C [41]. The DSC pattern of MTK nanocrystals suspended in aqueous vehicle and embedded hydrogels was quite analogous to that of the MTK free acid powder, with no other distinctive peaks. In TEM observation, MTK nanocrystals embedded in hydrogel was physically stable, with no changes in crystal size over 2 months (Data not shown). Taken together, we concluded that MTK free acid powder formed by acid-base neutralization was lucratively split below 200 nm, with no crystalline changes and/or polymorphic transition during ultra-sonication process and subsequent embedding process.

### 3.4. In Vitro Drug Release Profile from MTK Nanocrystal Suspension and Hydrogel

In vitro drug release profiles from MTK nanocrystal suspension or hydrogel were comparatively evaluated with those from the drug solution and conventional hydrogel, under sink condition. To ensure sufficient drug solubility in dissolution media, 0.5% *w/v* of sodium lauryl sulfate (SLS) was contained in 10 mM phosphate-buffered saline (pH 7.4). As control, drug solution was prepared by dissolving the weak acid compound in alkaline solution (10 mM NaOH, pH 11.2). Conventional hydrogel was formulated by adding xanthan gum (0.5 *w*/*v*%) to drug alkaline solution at the same concentration with the nanocrystal hydrogel.

Under the sink condition, nanocrystal suspension was completely dissolved and released at the first sampling time (5 min), exhibiting comparable dissolution profile with the drug alkaline solution (Figure 3). This rapid dissolution of nanocrystal suspension under sink condition can be explained by Noyes-Whitney equation; dM/dt = k∙S∙C_s_, where dM/dt, dissolution rate; k, rate constant; S, surface area of the drug particle; C_s_, drug solubility in dissolution media. The lessening in drug particle size led to a drastically increased surface area, thus enhancing the dissolution rate of hydrophobic compound in dissolution media [42]. The embedment of drug molecules or drug nanocrystals in the xanthan gum matrix markedly hindered drug release; the cumulative amount of drug released from hydrogels was gradually increased for 3 h, showing over 80% drug release. Although drug release from the nanocrystal hydrogel was slower than that of the conventional hydrogel due to the additional dissolution step prior to diffusion out process, the extent of drug released after 2 h was quite comparable with that of the conventional hydrogel.

### 3.5. Photo-Stability of MTK Nanocrystal Formulations

The chemical stability of MTK nanocrystal or conventional preparations was comparatively evaluated under light exposure, as the leukotriene antagonist was reported to be extremely susceptible to light, heat, and oxidative degradations. As previously reported, light exposure of drug alkaline solution resulted in marked degradation of MTK, showing over 55% decrease in residual drug content (Figure 4A). Correspondingly, the amounts of *cis*-isomer and sulfoxide in the drug solution were sharply increased by 10.5% and 2.9%, respectively, after 8 h. On the contrary, photo-induced degradation of MTK in the nanocrystal suspension was reduced as compared to that of the drug solution, preserving over 60% of the drug residue after 4 h. In addition, the amounts of *cis*-isomer and sulfoxide in the preparation were markedly decreased to 3.9% and 2.2%, respectively. This result could be explained considering the structural characteristics and the low drug solubility in the nanocrystal suspension. In the nanocrystal suspension, drug molecules present in the outer region of the nanocrystals and small fraction of the drug dissolved in aqueous solutions are exposed to light [25]. On the contrary, in the drug alkaline solution, the labile compound existed in the solution at a molecular level and thus, MTK molecules might be exposed to the UV–Vis rays more profoundly, causing extensive degradation into *cis*-isomer and sulfoxide in the aqueous solution.

On the other hand, embedding of drug nanocrystals or molecules into hydrogel matrix considerably improved the photo-stability of MTK. In case of nanocrystal hydrogel, providing over 70% of the drug remaining after 8 h, which is over 40% higher as compared to the nanocrystal suspension (Figure 4A). The extent of *cis*-isomer and sulfoxide content in the nanocrystal hydrogel were noticeably lowered to 2.9% and 1.5%, respectively. The formation of physical barrier around the drug nanocrystals diminished light transmission in preparation, alleviating light exposure to nanocrystal and/or drug molecules. The overall order of the chemical stability of MTK was as follows: nanocrystal hydrogel > conventional hydrogel > nanocrystal suspension > drug alkaline solution. These findings suggested that the formulation of MTK nanosuspension and further incorporation into the polymeric matrix were effective in stabilizing the labile leukotriene antagonist in the external preparation.

The photo-stability of MTK in the nanocrystal or conventional preparations was further evaluated in the presence of Sunset Yellow FCF, as photo-stabilizing agent. The employment of the excipients exhibiting similar absorption spectra with targeted therapeutic agent reduced the undesirable light exposure, hampering photo-degradation of the active compounds. As expected, the degradation of MTK was markedly retarded in all formulations, providing over 60% of the drug remaining after 8 h (Figure 4B). The MTK content of nanocrystal suspension and hydrogel was determined to approximately 76% and 81% in the presence of the UV/Vis absorber, respectively, which was markedly higher as compared to 49% and 70% of the formulations in the absence of Sunset Yellow FCF. Correspondingly, the contents of *cis*-isomer and the sulfoxide content in the MTK nanocrystal suspension with Sunset Yellow FCF were markedly decreased, resulting in less than 20% and 50% compared to the nanocrystal suspension with no photo-stabilizer. It is coincided with the previous report that the chemical stability of MTK in oral liquid syrup was markedly improved by adding the coloring agents, by absorbing light in the UV and visible region ranging from 350 to 500 cm^−1^ [43].

### 3.6. In Vivo Pharmacokinetic Profile after Topical Administration of MTK Nanocrystal Hydrogel in Rats

The plasma concentration-time profile of the MTK following topical administration of the nanocrystal or conventional hydrogels in dorsal skin of rats is depicted in Figure 5, and relevant pharmacokinetic parameters calculated from the pharmacokinetic profiles are presented in Table 3. There were no signs on their physical condition and behavior, morbidity, and mortality of rats during the pharmacokinetic study (data not shown). Drug dose topically administered to the skin was 25 mg/kg, which is lower than oral no observable adverse effect level (NOAEL) of MTK reported [44], thus, exhibiting no marked adverse effects following single topical administration. In long-term chronic toxicity test in rats and mice for 12 months, the NOAEL value was estimated 50 mg/kg [44].

In both groups, the plasma levels of the leukotriene antagonist were steeply elevated after topical application, reaching the maximum levels within 2 h. There was no significant difference in the *C*_max_ values between the nanocrystal and conventional hydrogel-treated groups, exhibiting *C*_max_ values of 5.9 and 5.3 ng/mL, respectively. Then, we observed that the MTK levels in the plasma fluctuated between 2 and 6 ng/mL 5 h post-dosing, probably due to a complementary process between absorption and metabolism and/or elimination. The plasma concentration–time profiles of MTK obtained from both groups were quite fluctuated with large deviations between individuals. As the leukotriene antagonist is extremely lipophilic compound (log *P* value of 8.79) [8], it might struggle easily crossing the skin layer. In general, small (molecular weight below 500 Da) compounds with appropriate lipophilicity (log *P* value between 1 and 3) and a low melting point are more suitable for skin penetration [45]. Moreover, MTK has been reported to be extensively metabolized by multiple cytochrome P450s such as CYP2C8 and CYP2C9 and CYP3A4 and glucuronidase including UGT1A3, which is also abundant in skin layer [46]. Thus, extensive metabolism of MTK molecules in the liver and even skin tissue might contribute large variations. Afterward, the drug concentration in the plasma was decreased to less than 1 ng/mL 12 h post-dosing, with the elimination *T*_1/2_ values between 9.7 h and 6.2 h. There was no significant difference in *AUC*_(0–24 h)_ values between nanocrystal and conventional hydrogel-treated groups (20.1 and 23.5 ng∙h/mL, respectively). Thus, no significant differences in the extent and rate of drug permeation after topical administration of nanocrystal or conventional hydrogels were correlated with the in vitro dissolution profile of MTK, as described above, denoting that nanocrystalized drug powder was rapidly dissolved and/or adsorbed in the relevant skin layer by forming a high concentration gradient between the hydrogel and the stratum corneum. Moreover, intact MTK nanocrystals and/or dissolved drug molecules might have been penetrated the systemic circulation through the surrounding follicular epithelium, exhibiting comparable plasma concentration profiles of conventional hydrogel. From these findings, we concluded that nanocrystal hydrogels could be a potent tool that can provide comparable skin permeability with markedly improved chemical stability of MTK.

## 4. Conclusions

A novel nanocrystal system of MTK was lucratively prepared by acid-base neutralization and ultra-sonication method. Drug nanocrystals stabilized by PVP K30 polymer were approximately 130 nm in size, globular shaped, and were in amorphous state. The incorporation of MTK nanocrystals into xanthan gum hydrogel did not alter the physical characteristics of the drug nanocrystals. The nanocrystal hydrogel exhibited markedly improved photo-stability as compared to the drug solution, conventional hydrogel, and even nanocrystal suspension, considering the aspects of drug remaining and the generation of two main degradation products, *cis*-isomer and sulfoxide. Moreover, there was no noticeable difference in the plasma concentration-time profile between the nanocrystal and the conventional hydrogels, exhibiting equivalent AUC and *C*_max_ values. Therefore, the novel nanocrystal hydrogel represents a promising tool for transdermal delivery of MTK, a poorly soluble labile compound, improving patient compliance, especially in children and elderly.

## Figures and Tables

**Figure 1 pharmaceutics-12-00018-f001:**
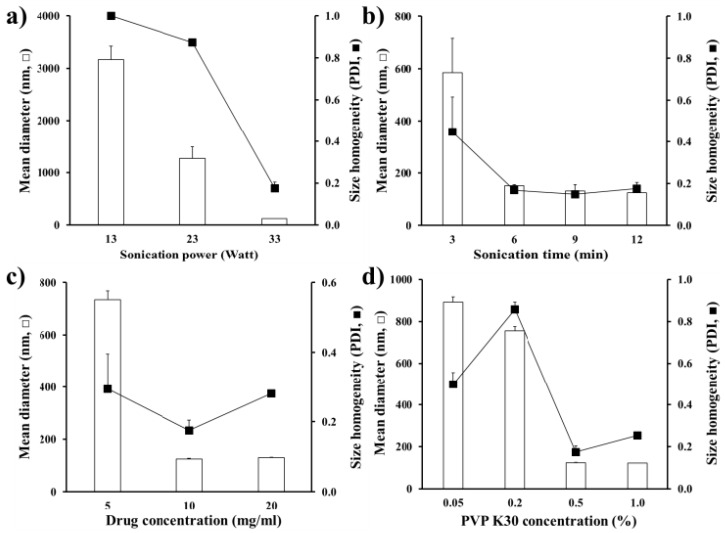
Effect of process variables (**a**, ultra-sonication power; **b**, sonication time; **c**, drug concentration; **d**, PVP K30 concentration in the nanosuspension) on the median diameter and homogeneity of MTK nanocrystals fabricated by acid-base neutralization and ultra-sonication method. Note: (**a**) The sonication time, drug concentration, and PVP K30 concentration were fixed to 6 min, 10 mg/mL, and 0.5 *w*/*v*%, respectively. (**b**) The sonication powder, drug concentration, and PVP K30 concentration were fixed to 33 Watts, 10 mg/mL, and 0.5 *w*/*v*%, respectively. (**c**) The sonication powder, sonication time, and PVP K30 concentration were fixed to 33 Watts, 6 min, and 0.5 *w*/*v*%, respectively. (**d**) The sonication powder, sonication time, and drug concentration were fixed to 33 Watts, 6 min, and 10 mg/mL, respectively. Three batches of each sample were prepared, and the data are presented as mean ± SD (*n* = 3).

**Figure 2 pharmaceutics-12-00018-f002:**
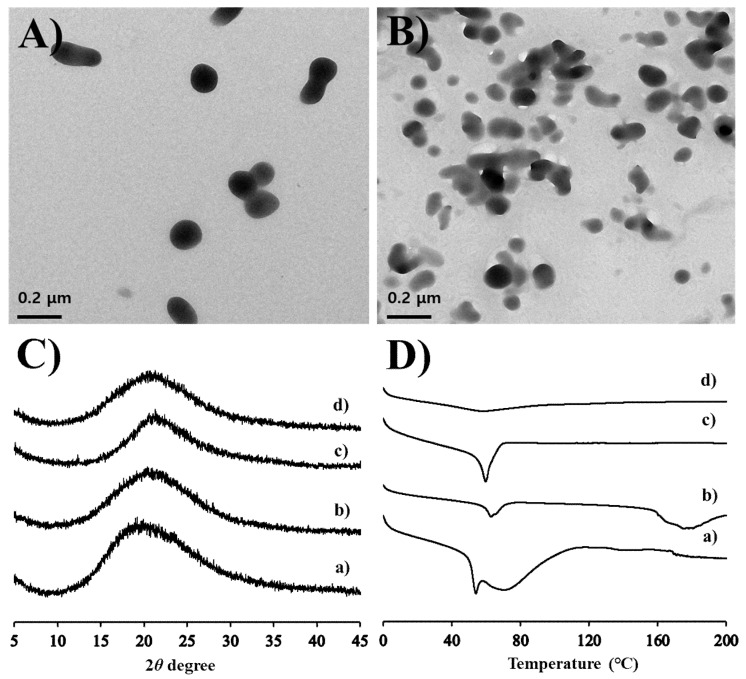
Morphological and physical characteristics of the MTK nanocrystal suspension and the hydrogel. SEM (scanning electron microscope) images of the MTK nanocrystals (**A**) suspended in aqueous suspension or (**B**) embedded in the hydrogel. (**C**) XRD (X-ray diffraction) patterns and (**D**) DSC (differential scanning calorimeter) curves of (a) MTK-Na powder, (b) MTK free-acid powder, (c) MTK nanocrystal suspension, and (d) MTK nanocrystal hydrogel.

**Figure 3 pharmaceutics-12-00018-f003:**
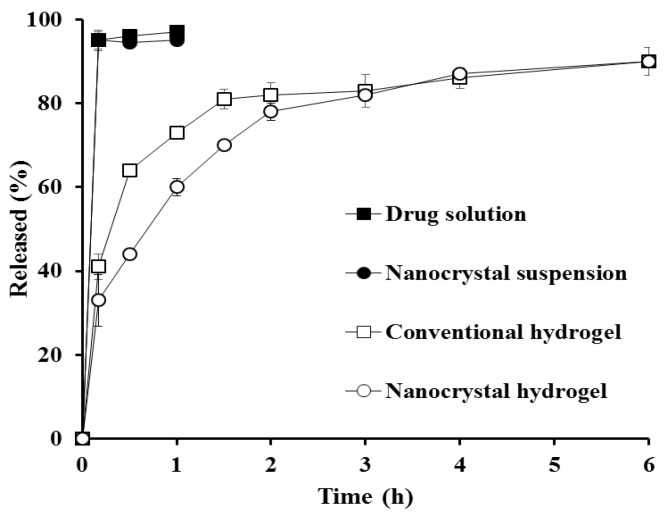
In vitro dissolution profiles of MTK from nanocrystal or conventional formulations under sink condition. Note: The sink condition was continued by adding 0.5% *w*/*v* of sodium lauryl sulfate (SLS) to 10 mM phosphate-buffered saline. Released (%) was calculated by dividing the amount of drug dissolved at determined time by the amount of drug loaded, and then multiplying by 100. The data are expressed as mean value (*n* = 3) and the SD is expressed as error bar on each point.

**Figure 4 pharmaceutics-12-00018-f004:**
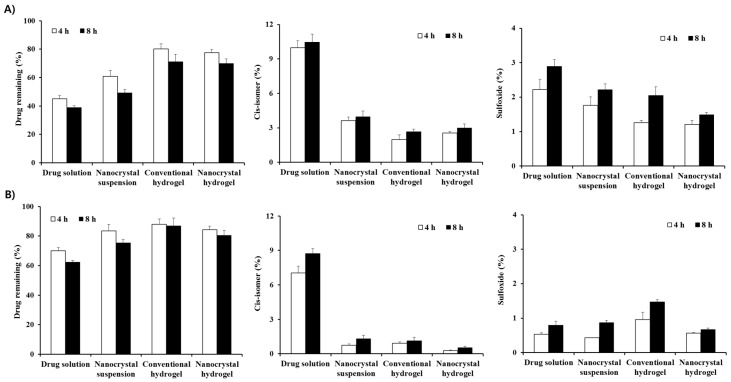
Drug remaining, and formation of *cis*-isomer and sulfoxide in the MTK nanocrystal or conventional formulations under light exposure, (**A**) in the absence of Sunset Yellow FCF and (**B**) in the presence of the photo-stabilizer. The data is represented as mean ± SD (*n* = 3).

**Figure 5 pharmaceutics-12-00018-f005:**
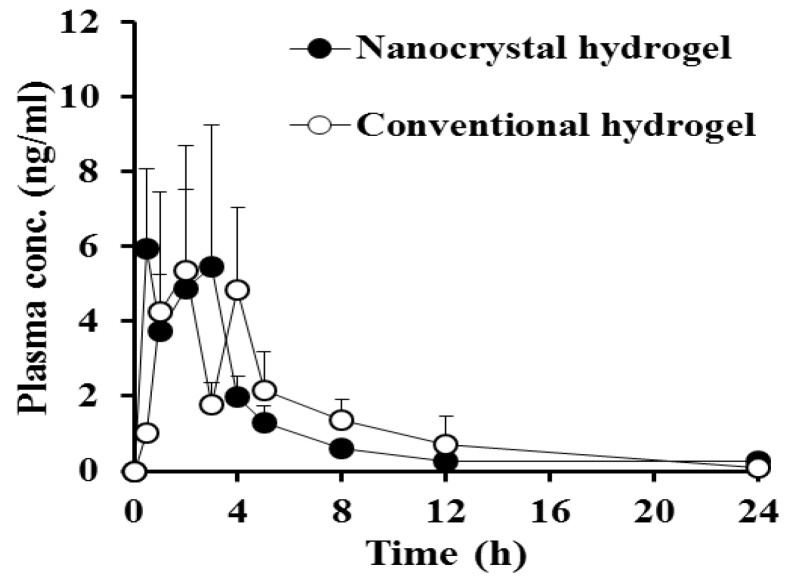
Plasma concentration–time profiles of MTK following single administration of the nanocrystal hydrogel or conventional hydrogel on the dorsal skin of healthy rats at a dose of 15 mg/kg. Each point represents mean ± standard error (*n* = 6).

**Table 1 pharmaceutics-12-00018-t001:** Effect of different types of stabilizers on the crystal size, homogeneity, and dispersibility of MTK nanocrystals in the aqueous vehicle.

Stabilizer (% *w*/*v*) ^1^	Crystal Size (nm) ^2^	Homogeneity (PDI) ^2,3^	Dispersibility ^4^
- ^5^	1614.3 ± 402.5	0.961 ± 0.067	Aggregated
PVP K30	129.7 ± 1.1	0.281 ± 0.007	Re-dispersible
Kollidon VA64	587.1 ± 177.7	0.556 ± 0.119	Re-dispersible
HPMC-2910	319.3 ± 2.1	0.308 ± 0.065	Aggregated
Poloxamer-188	89.9 ± 0.3	0.269 ± 0.006	Aggregated
Poloxamer-407	100.6 ± 1.0	0.353 ± 0.046	Aggregated
Tween 20	89.9 ± 0.4	0.290 ± 0.002	Aggregated
Tween 80	97.6 ± 0.6	0.275 ± 0.011	Aggregated
Kolliphor RH40	97.9 ± 0.6	0.451 ± 0.007	Aggregated
Cremophor EL	117.6 ± 0.5	0.184 ± 0.006	Aggregated
Solutol HS15	145.7 ± 0.8	0.251 ± 0.009	Aggregated

^1^ The concentration of stabilizers in the formulation was set to 0.5% *w*/*v*. ^2^ Data are expressed as mean ± SD (*n* = 3). ^3^ Polydispersity index, calculated by dividing weight average molecular weight by number average molecular weight. ^4^ Visually assessed after a week-long storage at 40 °C. ^5^ Prepared with no stabilizer.

**Table 2 pharmaceutics-12-00018-t002:** Physicochemical characteristics of the optimized MTK nanocrystal suspension.

Parameters	MTK Nanocrystal Suspension
MTK concentration (mg/mL)	10.93 ± 0.23
Suspended (mg/mL)	10.88 ± 0.20
Dissolved (mg/mL)	0.05 ± 0.01
Particle size (nm)	102.3 ± 3.0
PDI	0.238 ± 0.056
Zeta potential (mV)	−3.6 ± 0.7
pH	4.1 ± 0.1

Data are expressed as mean ± SD (*n* = 3).

**Table 3 pharmaceutics-12-00018-t003:** Pharmacokinetic parameters of MTK following dorsal application of the nanocrystal hydrogel or conventional hydrogel in rats.

Parameters	Nanocrystal Hydrogel	Conventional Hydrogel
AUC_(0–24 h)_ (ng·h/mL)	20.1 ± 5.2	23.5 ± 7.0
AUC_(0–inf)_ (ng·h/mL)	20.8 ± 5.7	26.6 ± 8.1
*C*_max_ (ng/mL)	5.9 ± 2.1	5.3 ± 2.2
*T*_max_ (h)	0.5	2.0
t_1/2_ (h)	9.7 ± 3.3	6.2 ± 5.4

Note: Data are expressed as mean ± standard error (*n* = 6). Abbreviations: AUC_0–__24 h)_, area under the plasma concentration–time curve from zero to 24 h; AUC_(0–inf)__,_ area under the plasma concentration–time curve to infinite time; C_max_, maximum plasma concentration; T_max_, time to reach maximum plasma concentration; T_1/2_, elimination half-life of the drug.

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
