# Peer review of "Montelukast Nanocrystals for Transdermal Delivery with Improved Chemical Stability"

_pharmaceutics, 2019, doi:10.3390/pharmaceutics12010018_

Round 1

Reviewer 1 Report

The manuscript addresses an interesting question and is generally well structured. Despite an overall good English writing, the manuscript should be revised before publication to correct minor typos, the excessive use of the article ‘the’ and the use of certain words/expressions that are not usually seen in scientific publications (e.g. ‘width of the size distribution’, ‘tap water and standardized chow’, ‘scrutinized’, ‘lucratively split’, etc.). In further detail, I would like the authors to take into consideration the following comments:

Major comments:

The authors used a selection of excipients to stabilize the nanocrystals – how was this choice made? Are there any previous studies published or performed to support the use of these specific compounds and not others? In lines 278-281, the authors hypothesize the possible manner in which the excipients contribute to the nanocrystals’ stability – this must be supported by the appropriate literature references. Why was the concentration of stabilizers set to 0.5% w/v, had this been published elsewhere or screened beforehand? In lines 294-296, the authors state the effect of increasing sonication intensity in the nanocrystals’ size was “expected” – if so, this must be supported by the appropriate literature references. In section 3.2., it is described how the authors screened 4 variables in terms of their effect on nanocrystals’ size; however, this is not a ‘multi-variable’ screening study as an appropriate experimental design would be – how did the authors know at which value to fix each parameter? What do the authors mean by “the particle size of novel nanocrystal system was supposed to be appropriate for transdermal delivery”? This statement must be supported by the appropriate literature references. When the authors mention the ‘Noyes-Whitney equation’ this should be indicated. In lines 391-393, it is mentioned that a decrease in particle size leads to an increased surface area, hence facilitating the “dissolution rate of hydrophobic compound” – is this compound actually dissolved in the media or rather dispersed due to its hydrophobicity? Why did the authors not include any control group in the in vivo study? For example, an alternative administration route would be interesting to compare the performance of the topical formulations, since none of the tested formulations is currently being used in the market. When discussing the in vivo results, the authors mention the nanocrystals could be forming a reservoir in a deeper skin layer, however, that is not supported by the results themselves – wouldn’t a reservoir lead to a more sustained and slow release of the drug?

Minor comments:

In line 27, “three-tenths and a half” is not an appropriate way to write an amount. In line 45, “age group of 624 months” is either a typo or should be corrected to the correct number in years. In line 64, the authors should replace “penetrate the hair follicles” for “penetrate the skin through the hair follicles”. Lines 65-66 should be rephrased, as they are confusing. In line 115, authors should clarify in which conditions the gels were incubated. In line 134, what does the word “brewed” mean in this context? In line 156, the expression “to colonize the nanocrystals in the aqueous vehicle” is not scientific. Why was acetonitrile used as drug solvent for in vitro dissolution profiles and methanol for drug content studies? “Food Yellow 5” should be replaced by “tartrazine” which is the appropriate compound name. “Normal rats” should be replaced by “healthy rats”. “Ball-milling” should be replaced by “bead-milling”. In figure 1, concentration of PVP K30 should be in % as in previous data. Figure 2B could have better resolution/quality. In line 468, “a-fifth” and “a-half” should be replaced by 20% and 50% respectively.

Author Response

The authors used a selection of excipients to stabilize the nanocrystals – how was this choice made? Are there any previous studies published or performed to support the use of these specific compounds and not others?

→ Thanks for your invaluable comment. To select appropriate stabilizer in designing MTK nanocrystal suspension, different stabilizers including hydrophilic polymers and surfactants were screened and was evaluated by the aspects of crystal size, homogeneity, and dispersibility (Table 1). Among them, PVP K30 polymer markedly decreased the crystal size below 150 nm, and were physically stable with excellent re-dispersibility in the vehicle. Thus, PVP K30 polymer was selected for preparation of MTK nanocrystal suspension. Data on the selection of stabilizer are represented in Table 1, and no other preliminary and/ or published data are available. Thanks again for your comment.

In lines 278-281, the authors hypothesize the possible manner in which the excipients contribute to the nanocrystals’ stability – this must be supported by the appropriate literature references.

→ Thanks for your invaluable comment. In accordance to your comment, appropriate references were attached as follows (see page 6, line 284): “These hydrophilic hydrogen-bonding acceptor polymers might be adsorbed on the surface of MTK nanocrystals by hydrogen bond and/or van der Waals interaction, and predominantly contribute to disperse the hydrophobic nanocrystals in the aqueous media, with no aggregation [36,37].”

Taylor, L.S.; Zografi, G. Spectroscopic characterization of interactions between PVP and indomethacin in amorphous molecular dispersions. Pharm. Res. 1997, 14, 1691–1698. Wen, H.; Morris, K.R.; Park, K. Study on the interactions between polyvinylpyrrolidone (PVP) and acetaminophen crystals: partial dissolution pattern change. J. Pharm. Sci. 2005, 94, 2166–2174.

Why was the concentration of stabilizers set to 0.5% w/v, had this been published elsewhere or screened beforehand? In lines 294-296, the authors state the effect of increasing sonication intensity in the nanocrystals’ size was “expected” – if so, this must be supported by the appropriate literature references.

→ Thanks for your invaluable comment. The concentration of stabilizer for screening test was set to 0.5% w/v through preliminary experiment. This is described in the manuscript as follows (See page 6, line 273): “The concentration of stabilizer for screening test was set to 0.5% w/v through preliminary experiment (data now shown).”Also, in accordance to your comment, appropriate reference was attached as follows (see page 7, line 302): “As expected [38]”.

Ho, M.J.; Lee, D.R.; Im, S.H.; Yoon, J.A.; Shin, C.Y.; Kim, H.J.; Jang, S.W.; Choi, Y.W.; Han, Y.T.; Kang, M.J. Design and in vivo evaluation of entecavir-3-palmitate microcrystals for subcutaneous sustained delivery. Eur. J. Pharm. Biopharm. 2018, 130, 143–151.

In section 3.2., it is described how the authors screened 4 variables in terms of their effect on nanocrystals’ size; however, this is not a ‘multi-variable’ screening study as an appropriate experimental design would be – how did the authors know at which value to fix each parameter?

→ Thanks for your invaluable comment. Four parameters to control the crystal size were set through previous report and our preliminary experiment. Also, in accordance to your comment, appropriate references were attached as follows (see page 7, line 301): “Four formulation variables were set from the previous report [38] and our preliminary experiment.”

Ho, M.J.; Lee, D.R.; Im, S.H.; Yoon, J.A.; Shin, C.Y.; Kim, H.J.; Jang, S.W.; Choi, Y.W.; Han, Y.T.; Kang, M.J. Design and in vivo evaluation of entecavir-3-palmitate microcrystals for subcutaneous sustained delivery. Eur. J. Pharm. Biopharm. 2018, 130, 143–151.

What do the authors mean by “the particle size of novel nanocrystal system was supposed to be appropriate for transdermal delivery”? This statement must be supported by the appropriate literature references.

→ Thanks for your invaluable comment. In accordance to your comment, appropriate references were attached as follows (see page 8, line 337): “The particle size of novel nanocrystal system was supposed to be appropriate for transdermal delivery [23,39,40].”

Patzelt,; Richter, H.; Knorr, F.; Schäfer, U.; Lehr, C.M.; Dähne, L.; Sterry, W.; Lademann, J. Selective follicular targeting by modification of the particle sizes. J. Control. Release 2011, 150, 45–48. Pireddu, R,; Caddeo, C,; Valenti, D,; Marongiu, F,; Scano, A,; Ennas, G,; Lai, F,; Fadda, A.M,; Sinico, C. Diclofenac acid nanocrystals as an effective strategy to reduce in vivo skin inflammation by improving dermal drug bioavailability. Colloids Surf. B Biointerfaces 2016, 143, 64– Li,; Wang, D.; Lu, S.; Zeng, L.; Wang, Y.; Song, W.; Liu, J. Pramipexole nanocrystals for transdermal permeation: Characterization and its enhancement micro-mechanism. Eur. J. Pharm. Sci. 2018, 124, 80–88.

When the authors mention the ‘Noyes-Whitney equation’ this should be indicated.

→ Thanks for your invaluable comment. In accordance to your comment, Noyes-Whitney equation was indicated in the manuscript as follows (See page 10, line 405): “This rapid dissolution of nanocrystal suspension under sink condition can be explained by Noyes-Whitney equation; dM/dt = k∙S∙Cs, where dM/dt, dissolution rate; k, rate constant; S, surface area of the drug particle; Cs, drug solubility in dissolution media.”

In lines 391-393, it is mentioned that a decrease in particle size leads to an increased surface area, hence facilitating the “dissolution rate of hydrophobic compound” – is this compound actually dissolved in the media or rather dispersed due to its hydrophobicity?

→ Thanks for your critical comment. As you mentioned, nano-sized drug might be presented in the medium in dissolved state or dispersed state. After sampling, we centrifuged the sample to remove the undissolved, dispersed drug crystals and quantitated the only dissolved drug. The protocol in detail was described in the Materials and Methods section (See page 4, line 176).

Why did the authors not include any control group in the in vivo study? For example, an alternative administration route would be interesting to compare the performance of the topical formulations, since none of the tested formulations is currently being used in the market.

→ Thanks for your invaluable comment. We absolutely agree that it would be better to include a marketed product-treated control group. However, the main purpose of this study was to comparatively evaluate the chemical stability and skin permeability of nanocrystals with the drug solution, rather than comparing and replacing the current treatment regimen. Nevertheless, we agree on your opinion and appreciate your comment and your understanding.

When discussing the in vivo results, the authors mention the nanocrystals could be forming a reservoir in a deeper skin layer, however, that is not supported by the results themselves – wouldn’t a reservoir lead to a more sustained and slow release of the drug?

→ Thanks for your invaluable comment. In our permeation study, a thin film made from polyurethane and acrylamide (Tegaderm®, 10 x 12 cm) was attached onto the application site in both nanocrystal- and conventional hydrogels-treated groups, to avoid the drying of the nanocrystal or conventional hydrogels. This pretreatment might be diminished the differences in permeation profiles between nanocrystal- and conventional hydrogels. The sentence was removed from the manuscript, because it could not be backed up by the test result. Thanks again for your comment.

In line 27, “three-tenths and a half” is not an appropriate way to write an amount.

→ Thanks for your kind review. In accordance to your comment, the manuscript was corrected as follows (see page 1, line 26): “yielding only 30% and 50% amount of cis-isomer and sulfoxide as the major degradation products, as compared to those of drug alkaline solution.”

In line 45, “age group of 624 months” is either a typo or should be corrected to the correct number in years.

→ Very sorry for our typo and thanks for your kind review. The manuscript was correct as follows (see page 1, line 44): “age group of 6-24 months”

In line 64, the authors should replace “penetrate the hair follicles” for “penetrate the skin through the hair follicles”.

→ Thanks for your kind review. In accordance to your comment, “penetrate the hair follicles” was corrected to “penetrate the skin through the hair follicles” (see page 2, line 63).

Lines 65-66 should be rephrased, as they are confusing.

→ Thanks for your kind review. The sentence was rephrased in the manuscript as follows (See page 2, line 64): “Moreover, drug crystal system can provide higher chemical stability as compared to the conventional drug-solubilized formulations by minimizing the exposure to light, oxygen, and moisture at the molecular level [24].”

In line 115, authors should clarify in which conditions the gels were incubated.

→ Thanks for your kind review. The sentence was clarified as follows (see page 3, line 114): “Prepared hydrogels were incubated overnight at room temperature to remove the air bubbles.”

In line 134, what does the word “brewed” mean in this context? In line 156, the expression “to colonize the nanocrystals in the aqueous vehicle” is not scientific.

→ Thanks for your kind review and sorry for typos. In accordance to your comment, these sentences in the manuscript were corrected as follows : (see page 3, line 135): “Approximately 50 μl of sample was loaded onto the copper grid and was gently blown up for 20 min, to diminish aqueous vehicle.” Also see page 4, line 157: “Nanocrystal suspension (1 ml) was ultra-centrifuged at 13,000 rpm for 10 min to settle the nanocrystals in the aqueous vehicle.”

Why was acetonitrile used as drug solvent for in vitro dissolution profiles and methanol for drug content studies?

→ Thanks for your kind review. Actually, there was no significant difference between the two organic solvents. But, the active compound was more stable in acetonitrile, so the dissolution test samples that took a long time to be analyzed were diluted with acetonitrile.

“Food Yellow 5” should be replaced by “tartrazine” which is the appropriate compound name.

→ Thanks for your kind review and sorry for confusing. The colorant used in our study was “Sunset Yellow FCF (C16H10N2Na2O7S2)”, which is named in Food Yellow No.5 in South Korea, FD&C Yellow 6 in the United States, and E Number E110 in Europe. In accordance to your comment, “Food Yellow 5” was corrected to “Sunset Yellow FCF” in the manuscript to clearly describe the photo-stabilizer. Thanks again for your kind comment.

“Normal rats” should be replaced by “healthy rats”.

→ Thanks for your kind review. In accordance to your comment, “Normal rats” was corrected to “healthy rats” in the manuscript.

“Ball-milling” should be replaced by “bead-milling”.

→ Thanks for your kind review. In accordance to your comment, “ball-milling” was corrected to “bead-milling” in the manuscript (see page 6, lone 262).

In figure 1, concentration of PVP K30 should be in % as in previous data.

→ Thanks for your kind review. In accordance to your comment, the concentration of PVPK30 was identically expressed to w/v%, not mg/ml” (see page 8, line 322).

Figure 2B could have better resolution/quality.

→ Thanks for your kind review. Figure 2B is the TEM image of the nanocrystals embedded into the xanthan gum. As the drug nanocrystals were located in the matrix, it was quite difficult to obtain a more focused image. Nevertheless, we have confirmed that spherical shape of nanocrystals was preserved, which are similar to that of nanocrystal suspension. Figure 2B is the highest quality image we have and unfortunately, it is now difficult to provide more high-resolution images with difficulty in further experiment. Thanks again for you comment and understanding.

In line 468, “a-fifth” and “a-half” should be replaced by 20% and 50% respectively.

→ Thanks for your kind review. In accordance to your comment, “a-fifth” and “a-half” were replaced by 20% and 50%, respectively (see page 12, line 483).

Reviewer 2 Report

     Im et al. designed a novel nanocrystal system of montelukast (MTK) to improve the transdermal delivery while ensuring chemical stability of the labile compound. MTK nanocrystal suspension was fabricated using acid-base neutralization and ultra-sonication technique and was characterized. The in vitro drug release profile from the nanocrystal hydrogel was comparable to that of the conventional hydrogel because of the rapid dissolution pattern of the drug nanocrystals. Additionally, no marked pharmacokinetic difference between the nanocrystal and the conventional hydrogels, exhibiting equivalent extent and rate of drug absorption after topical administration was found in animal studies. Finally, they claimed that this MTK nanocrystal formulation is promising tool to deliver MTK for the treatment of chronic asthma or seasonal allergies. Below are my comments:

In the 2.7 In vivo transdermal delivery of MTK nanocrystal formulations subsection:

(1) Please provide ethics statement for the animal study at Dankook  University (Cheonan, Korea) including approval number and approval date.

(2) For safety test in the animal (rat) study, please provide the material and method, and results in the related section, respectively.

As the MTK nanocrystal formulations is the main points in this article, the authors should add the data for its characteristics such as protein contents, size distribution, zeta potential and particle stabilities, etc. This analysis for MTK nanocrystal formulations should be embedded in the result section. In the abstract, the authors claimed “This novel nanocrystal system can be a potent tool for transdermal delivery of MTK in the treatment of chronic asthma or seasonal allergies, with better patient compliance, especially in children and elderly.” Please also have an echo to explain and confirm this statement in the conclusion section.

Author Response

In the 2.7 In vivo transdermal delivery of MTK nanocrystal formulations subsection: Please provide ethics statement for the animal study at Dankook  University (Cheonan, Korea) including approval number and approval date. For safety test in the animal (rat) study, please provide the material and method, and results in the related section, respectively.

→ Thanks for your invaluable comment. In accordance to your comment, ethics statement for the animal study at Dankook University (Cheonan, Korea) including approval number and approval date was described in the manuscript as follows (See page 5, line 203): “In vivo pharmacokinetic study of MTK-loaded nanocrystal preparation was carried out in the healthy rats, after approval from Institutional Animal Care and Use Committee (IACUC) of Dankook University (Cheonan, Korea) (DKU-19-032, approved data : Oct 8, 2019). Also, description in the safety of animals during the test was further added in the manuscript as follows (see page 5, line 214): “After 6 h post-administration, rats were allowed to freely access to water and standardized chow. There were no remarkable changes in general appearance and deaths throughout the pharmacokinetic study.”

As the MTK nanocrystal formulations is the main points in this article, the authors should add the data for its characteristics such as protein contents, size distribution, zeta potential and particle stabilities, etc. This analysis for MTK nanocrystal formulations should be embedded in the result section.

→ Thanks for your invaluable comment. In accordance to your comment, the physicochemical characteristics of the MTK nanocrystal formulation was further described in the manuscript.

1) In material and methods section (See page 3, line 128): “The zeta potential of the MTK nanocrystals (approximately pH 4.0) was also estimated using a Zetasizer Nano at 25°C. Samples (100 μl) were loaded into the capillary cell after 10-fold dilution with distilled water, and twenty runs were performed for each measurement. All measurements were carried out in triplicates at 25°C.

2) Also see page 8, line 336: “The particle size analysis also revealed that MTK nanocrystals with median diameter of 102.3 nm were effectively prepared with PDI value < 0.3 (Table 2).

3) Also see page 9, line 372: “Drug content analysis in suspension revealed that over 99.5% of MTK was suspended as solid-state in the formations, due to poor solubility of MTK, a weak-acid compound, in acidic environment (pH 4.1) (Table 2). The zeta potential of drug nanocrystals stabilized with PVP K30 polymer was neutral (-3.6 mV) (Table 2).

4) Also see page 9, line 370: Table 2. Physicochemical characteristics of the optimized MTK nanocrystal suspension.

Parameters

MTK nanocrystal suspension

MTK conc. (mg/ml)

10.93 ± 0.23

  Suspended (mg/ml)

10.88 ± 0.20

  Dissolved (mg/ml)

0.05 ± 0.01

Particle size (nm)

102.3 ± 3.0

PDI

0.238 ± 0.056

Zeta potential (mV)

-3.6 ± 0.7

pH

4.1 ± 0.1

Data are expressed as mean ± SD (n=3).

Also see page 10, line 389:In TEM observation, MTK nanocrystals embedded in hydrogel was physically stable, with no changes in crystal size over 2 months (Data not shown).”

In the abstract, the authors claimed “This novel nanocrystal system can be a potent tool for transdermal delivery of MTK in the treatment of chronic asthma or seasonal allergies, with better patient compliance, especially in children and elderly.” Please also have an echo to explain and confirm this statement in the conclusion section.

→ Thanks for your invaluable comment. In accordance to your comment, the significance and/or goal of the study was emphasized again in Conclusion section, as follows (See page 13, line 552): “Therefore, the novel nanocrystal hydrogel represents a promising tool for transdermal delivery of MTK, a poorly soluble labile compound, improving patient compliance, especially in children and elderly.”

Reviewer 3 Report

The manuscript presented here shows some advantageous results which would be useful in the current transdermal research remit. The authors needs to address a couple of minor points before the article can be deemed suitable for publication:

- A thorough read through is necessary with respect to the written English. Please ensure the English (i.e. punctuation and grammar is to of a good standard).

- Line 88 seems to include parenthesis without any content -

- Was the differential scanning calorimeter calibrated?

- Table 1: The first stabiliser is unclear.

- some physical stability testing would be beneficial here; perhaps FTIR to confirm structural stability

- Figure 2: d) label with critical peak/temp.

- Figure 3: Y axis needs to be more concise; what is being released?

- Also please comment on the validity of the results; overlapping error bars on figure 3 and figure 5.

Round 2

Reviewer 2 Report

For safety tests in the animal (rat) study, please provide methods (e.g., observation time, rat number, rat behavior, etc.) in details for evaluating the animal health in the section of material and method, and provide data or observation results (e.g., body wight, number of sick, number of death, survival rate, etc.) in the section of result, respectively. Additionally, I do not think it enough to observe rats for only 6 hours post administration, because the in vitro dissolution profiles of MTK from nanocrystal or conventional formulation (hydrogel) is reaching the submit (over 80%) after 6 hours (Figure 3). It should need more than 6 hours to have safety surveillance post-administration. Please provide the full name of PDI when it appears in the manuscript the first time.

Author Response

Thanks for reviewing the manuscript in detail: Ms. Ref. No.: pharmaceutics-662343 titled “Montelukast nanocrystals for transdermal delivery with improved chemical stability”. The reviewer's comments were carefully studied and reflected in the revised manuscript. Please find the separate sheets attached which addresses, point-by-point, the issues raised by the reviewer.

For safety tests in the animal (rat) study, please provide methods (e.g., observation time, rat number, rat behavior, etc.) in details for evaluating the animal health in the section of material and method, and provide data or observation results (e.g., body weight, number of sick, number of death, survival rate, etc.) in the section of result, respectively. Additionally, I do not think it enough to observe rats for only 6 hours post administration, because the in vitro dissolution profiles of MTK from nanocrystal or conventional formulation (hydrogel) is reaching the submit (over 80%) after 6 hours (Figure 3). It should need more than 6 hours to have safety surveillance post-administration.

→ Thanks for your invaluable comment and sorry for our insufficient or confusing descriptions on the safety of the animals during pharmacokinetic study. Rats were observed for 24 h after topical application of MTK, not 6 h, and no remarkably signs on their physical condition and behavior, morbidity, and mortality of rats were observed for24 h. In accordance to your comment, method and result concerning the safety of animals during the pharmacokinetic study were further described in the manuscript as follows (see page 5, line 215): “During the experimental period (24 h), individual rats were monitored for their changes in skin and fur, behavior pattern, morbidity, and mortality.”

Also see (page 12, line 491): “There were no signs on their physical condition and behavior, morbidity, and mortality of rats during the pharmacokinetic study (data not shown). Drug dose topically administered to the skin was 25 mg/kg, which is lower than oral no observable adverse effect level (NOAEL) of MTK reported [44], thus exhibiting no marked adverse effects following single topical administration. In long-term chronic toxicity test in rats and mice for 12 months, the NOAEL value was estimated 50 mg/kg [44].”

Huang, X.; Aslanian, R. G. Case studies in modern drug discovery and development. Chapter 8. Discovery and development of montelukast (Singulair®), edited by Young, R.N. 2012, 154-195, John Wiley & Sons, Inc.

Please provide the full name of PDI when it appears in the manuscript the first time.

→ Thanks for your kind review. In accordance to your comment, the full name of PDI was described in the manuscript as follows (see page 3, line 125): “Average particle size and polydispersity index (PDI) value, a measure of the width of the size distribution of MTK nanocrystal suspension were determined using a Zetasizer Nano dynamic light scattering particle size analyzer (Marlvern Instrument, Worcestershire, UK).”

Round 3

Reviewer 2 Report

The manuscript has been significantly improved, so I recommend it to be published.